# Extracellular Enzyme Stoichiometry Reveals Soil Microbial Carbon and Phosphorus Limitations in the Yimeng Mountain Area, China

Lu Wang [1], Kun Li [1], Jianyao Guo [2], Xiumei Liu [3], Jinhui Gao [4], Liang Ma [1], Jinhui Wei [1], Min Lu [5] and Chuanrong Li [1,*]

1   Taishan Forest Ecosystem Research Station, Key Laboratory of State Forestry Administration for Silviculture of the Lower Yellow River, Shandong Agricultural University, Tai'an 271018, China; 2019010126@sdau.edu.cn (L.W.); kunli@sdau.edu.cn (K.L.); 2020120842@sdau.edu.cn (L.M.); 2019110455@sdau.edu.cn (J.W.)
2   Shandong Forestry Protection and Development Center, Jinan 250000, China; guojianyao@shandong.cn
3   Department of Ecological Engineering, Shanghai Environment College, Shanghai 200135, China; xiaomi8869@163.com
4   Biodiversity Research Center, Yichun Branch of Heilongjiang Academy of Forestry Sciences, Yihcun 153000, China; zhl737603@163.com
5   Landscape Architecture Research Center, Shandong Jianzhu University, Jinan 250101, China; lumin@sdjzu.edu.cn
*   Correspondence: chrli@sdau.edu.cn

**Abstract:** Soil extracellular enzymes are considered key components in ecosystem carbon and nutrient cycling, and analysing their stoichiometry is an effective way to reveal the resource limitations on soil microbial metabolism. In this study, the soil and litter of *Quercus acutissima* plots, *Pinus thunbergii* plots, *Quercus acutissima–Pinus thunbergii* mixed-plantation plots, herb plots, and shrub plots in the state-owned Dawa Forest Farm in the Yimeng Mountain area were studied. The total carbon (C), nitrogen (N), and phosphorus (P) contents of litter and the physical and chemical properties of soil were analyzed, along with the activities of four extracellular enzymes related to the soil C, N, and P cycle: β-1,4-glucosidase (BG), β-1,4-N-acetylglucosaminidase (NAG), L-leucine aminopeptidase (LAP), and acid phosphatase (AP). The extracellular enzyme stoichiometric model was used to study and compare the metabolic limitations of soil microorganisms in different plots, and the driving factors of microbial metabolic limitations were explored by redundancy and linear regression analyses. The results showed that the values of BG/(NAG + LAP) were all higher than 1, the values of (NAG + LAP)/AP all lower than 1, and the vector angles of the five plots all greater than 45°, which indicated that the soil microorganisms were relatively limited by C and P. Redundancy and linear regression analysis revealed that soil physical properties (e.g., soil moisture) and litter total C make greater contributions to soil extracellular enzymes and stoichiometry than the other investigated soil parameters, whereas soil chemical properties (e.g., soil organic C and available P) predominantly controlled vector properties. Therefore, microbial metabolism limitations are greatly regulated by soil physical and chemical properties and litter total C and N. Compared with the forest plots, the soil microbial C (1.67) and P (61.07°) limitations of herb plots were relatively higher, which means that the soil microbial communities of forest plots are more stable than those of herb plots in the Yimeng Mountain area. Forest plots were more conducive than other plots to the improvement of soil microbial ecology in this area. This study could be important for illuminating soil microbial metabolism and revealing soil nutrient cycling in the Yimeng Mountain area ecosystem of China.

**Keywords:** soil physical and chemical properties; litter nutrition; extracellular enzyme stoichiometry; microbial nutrient limitation; Yimeng Mountain area

## 1. Introduction

Soil microbial communities are major participants in the global biogeochemical cycle; they drives the cycling of carbon (C) and other nutrients and regulate the decomposition of organic matter by producing a variety of extracellular enzymes [1]. Therefore, extracellular enzymes can be used to assess microbial nutrient requirements and metabolic processes [2]. Extracellular enzymes participate in the absorption and utilization of C, N, P, and other elements in soil biochemical reactions and have functions in catalysis, decomposition, transformation, synthesis, and other processes [3]. The levels of extracellular enzymes can reflect the functional traits of soil microorganisms and characterize the potential of soil with respect to specific biochemical reactions, and the roles of these enzymes are limited by various environmental factors [4]. Differences in the soil microenvironment and tree species affect the levels of soil extracellular enzymes [5]. Therefore, the determination of soil extracellular enzymes is very important for analysing the metabolic functions of soil microorganisms [6].

Extracellular enzyme stoichiometry combines ecological metabolism and ecological stoichiometry theory to estimate the limitations of ecosystem microbial metabolism from the perspective of enzymes [7]. It can reveal the nutrient cycles of an ecosystem [8] and reflect the metabolic characteristics of soil microbial nutrient requirements and limitations [9]. Current extracellular enzyme stoichiometry studies are focussed mostly on four extracellular enzymes: β-1,4-glucosidase (BG), β-1,4-N-acetylglucosaminidase (NAG), L-leucine aminopeptidase (LAP), and acid phosphatase (AP). The soil C-acquiring enzyme BG can be used to catalyse the C cycle. The soil N-acquiring enzymes NAG and LAP are responsible for the decomposition of peptidoglycan and leucine. The soil P-acquiring enzyme AP can catalyse the mineralization of organophosphorus chemicals [10]. Studies have revealed that it affects the activity of soil extracellular enzymes by changing the availability of soil nutrients, which, in turn, affects extracellular enzyme stoichiometry [11]. Peng and Wang showed that an N:P or C:P of extracellular enzymes lower than 1 indicates P limitation. As soil depth increases, the enzyme C:P level gradually decreases [12]. Adding N can promote the activity of C-degrading enzymes and increase the extracellular enzyme level of C:P [13]. Therefore, studying the quantitative characteristics of soil extracellular enzymes under different soil conditions is very important for assessing the C, N, and P turnover of ecosystems.

Microbial metabolism restriction is commonly used for the evaluation of extracellular enzyme carrier models. Vector length and angle can be calculated based on the C:N and C:P values of extracellular enzymes to quantify the acquisition of microbial C, N, and P [14], and vector characterization of extracellular enzymes can be used to evaluate the nutritional requirements of microbes and clarify the relative restrictions of microbial metabolism. A vector angle <45° indicates microbial N limitation, a vector angle >45° indicates microbial P limitation, and a greater vector length indicates greater C limitation [11]. In grassland ecosystems, the vector angle was generally 62.7° to 71.1°, and soil microbial metabolism is mainly limited by P under nitrogen addition. Vector length and vector angle are significantly related to the moisture and available phosphorus of soil. With increasing soil moisture, microbial C limitation increases [15,16]. Vegetation traits affect the stoichiometry of extracellular enzymes [17]; for example, broad-leaved forests have stronger effects on the stoichiometry of soil extracellular enzymes than coniferous forests in the Qinghai–Tibet Plateau in China [18], and the extracellular enzyme stoichiometry of litter is related to that of soil [19]. Therefore, studying the characteristics of microbial metabolism in the Yimeng Mountain area, China, can help clarify the mechanisms of soil microbial C and nutrient cycling in this region and the responses of different factors to the restriction of microbial metabolism in the context of global climate change.

Forests account for the largest amount of C storage among terrestrial ecosystems. Improving the stability and ecological service functions of forest ecosystems and increasing forest utilization are effective ways to address current climate change. Forest soils in the Yimeng Mountain area, China, are generally poor, arid, and barren, nutrient cycling

is restricted, and the metabolism of microorganisms is blocked. These conditions affect C retention and ecosystem stability [20]. To date, research on the soil in the Yimeng Mountain area has been focussed mainly on the spatial distribution of soil thickness and the effects of soil hydrological characteristics on the ecosystem functions of soil and water conservation [21,22]. We know little about the soil environments of different communities, the relationships between litter inputs and extracellular enzymes, and the mechanisms of nutrient limitation of soil microbial metabolism. Therefore, studying the characteristics of soil and litter in this area can reveal not only the metabolic characteristics of soil microbial C and nutrients but also the potential responses of microbial metabolism limitations to influencing factors.

In this study, we investigated the nutrient limitations of microbial communities in different vegetation communities in the state-owned Dawa Forest Farm in the Yimeng Mountain area, China. We sought to determine the characteristics of soil microbial metabolisms and their influencing factors in the different communities. The hypotheses considered in this study are as follows: (H1) Soil microorganisms are limited by C and P levels. (H2) Soil microbial nutrient limitations with herbs are greater than in areas with forest plots [10]. (H3) The soil microbial limitations of communities are affected by changes in soil physical and chemical properties and litter nutrients.

## 2. Materials and Methods

### 2.1. Study Site

The sites were located in the state-owned Dawa Forest Farm of the Yimeng Mountain area, China (latitude 35°30′2.04″ to 35°30′24.43″ N and longitude 117°55′44.14″ to 117°56′3.36″) (Figure S1). The area has a warm temperate continental monsoon climate and a frost-free period of 191 d, and the mean annual temperature and precipitation are 12.8 °C and 600 mm, respectively. The rock is dominated by gneiss and brown forest soil is predominant.

Five communities were selected as study plots in October 2019: herb, shrub, *Quercus acutissima*, *Pinus thunbergii*, and *Quercus acutissima–Pinus thunbergii* mixed-plantation plots. The forest plots were 30-year-old plantations. The average soil thickness was approximately 10 cm in the herb plots, approximately 15 cm in the shrub plots, and approximately 20 cm in the forest plots. There was a medium-thickness soil layer of approximately 30 cm in the understorey and ravine areas of the forest plots. The main species of vegetation in this area are *Quercus acutissima*, *Pinus thunbergii*, *Vitex negundo*, *Cymbopogon goeringii*, *Achnatherum pekinense*, *Viola collina*, *Conyza canadensis,* and *Artemisia stechmanniana*.

### 2.2. Experimental Design and Soil Sampling

Three plots of $5 \times 5$ m were established in the herb community, three plots of $10 \times 10$ m were established in the shrub community, and three plots of $30 \times 30$ m were established in each of the *Quercus acutissima*, *Pinus thunbergii*, and mixed-plantation communities, respectively. The spacing between adjacent plots were at least 10 m. In each plot, soil samples were collected using a soil auger (diameter, 4 cm) from the 0–15 cm soil layer at 15 random points and then mixed into a composite sample as one replicate; all the litter above the sampling points was collected [11]. Samples were collected form a total of 225 random points across the different sites, with three plots per community as three independent replicates. In total, 15 composite samples were established [11]. A subsample of each composite sample was immediately placed in an ice box, transported to the laboratory, and then stored at 4 °C for the analysis of extracellular enzyme activities within two weeks. The other subsample was air-dried for physicochemical analysis. The litter was brought to the laboratory and dried for the determination and analysis of C, N, and P contents.

### 2.3. Soil Physiochemical Analysis

Soil moisture was determined by oven-drying fresh soil at 105 °C for 24 h. Soil pH was measured at a soil-to-water ratio of 1:5 with a glass-electrode meter (FiveEsay Plus, Mettler Toledo, Schwerzenbach, Switzerland). Soil ammonia nitrogen and nitrate nitro-

gen ($NH_4^+$-N, $NO_3^-$-N) were measured by KCl extraction and continuous flow analysis (AA3, Bran+Luebbe, Norderstedt, Germany). Organic C was determined by the potassium dichromate–concentrated sulfuric acid titration method (Top Burret M, Eppendorf AG, Hamburg, Germany) [23]. Available P was determined by ammonium fluoride–hydrochloric acid extraction and concentrated sulfuric acid digestion, and molybdenum antimony was determined by a colorimetric method (UV2300, Hitachi, Shanghai, China) [24]. The litter samples were dried at 105 °C for 30 min and then at 65 °C to constant weight. Each dried sample was passed through a 100-mesh screen, and the total C ($TC_{litter}$) and N ($TN_{litter}$) contents were detected by an element analyzer (ECS4010, Costech, Milan, Italy). The total P ($TP_{litter}$) content was determined by the concentrated $H_2SO_4$–$H_2O_2$ digestion method followed by the molybdenum–antimony colorimetric method (UV2300, Hitachi, Shanghai, China) [25].

### 2.4. Assays of Extracellular Enzyme Activities

The activities of BG, NAG, LAP, and AP were measured fluorometrically using black 96-well microplates. Fresh soil samples were extracted with 125 mL of sodium acetate buffer (pH = 5) under continuous stirring with a magnetic stirrer and mixed at high speed into a slurry; the soil suspension (slurry) was continuously stirred as 200 μL aliquots were dispensed into the microplate wells that served as the sample assay. The blank and quench standard (slurry + standard), BG, NAG, and AP were measured using 4-methylumbelliferyl β-D-glucopyranoside, 4-Methylumbelliferyl-2-acetamido-2-deoxy-β-D-glucopyranoside, and 4-Methylumbelliferyl phosphate as substrates, respectively; 4-methylumbelliferone (MUB) was used as a standard. LAP was measured using L-Leucine-7-amido-4-methylcoumarin hydrochloride as a substrate; 7-amino-4-methylcoumarin (AMC) was used as the standard solution. The prepared plates were incubated at 20 °C in the dark following substrate addition, and fluorescence was measured using a microplate reader (Synergy MX, BioTek, Vermont, USA) at λ365 nm excitation and λ450 nm emission. The enzyme activities were expressed as nanomoles of substrate released per hour per gram of dry soil (nmol $g^{-1}$ $h^{-1}$) [26].

### 2.5. Extracellular Enzyme Stoichiometry Model

Two common methods using stoichiometric indicators were used to estimate soil microorganism limitation: plotting enzyme ratios and calculating vector lengths and angles of enzymes. Microbial resource limitations were judged based on a scatter plot of (LAP + NAG)/AP and BG/(LAP + NAG), with the four quadrants representing soil microbial N limitation, P limitation, C and N limitation, and C and P limitation. This method requires the plotting of (LAP + NAG)/AP on the *x*-axis and BG/(LAP + NAG) on the *y*-axis and denoting the horizontal and vertical coordinates of 1 as baselines [27,28].

All extracellular enzymes were logarithmically transformed. The soil extracellular enzyme C/N was expressed as lnBG/ln(NAG + LAP), the soil extracellular enzyme C/P was expressed as lnBG/lnAP, and the soil extracellular enzyme N/P was expressed as ln(NAG + LAP)/lnAP. Microbial metabolic limitations were quantified by the vector lengths and angles of extracellular enzymes [14]. Vector length, representing C limitation, was calculated as the square root of the sum of (lnBG/ln(NAG + LAP))$^2$ and (lnBG/lnAP)$^2$ (Equation (1)). Vector angle, representing N or P limitation, was calculated as the arctangent of the line extending from the plot origin to point (lnBG/lnAP, lnBG/ln(NAG + LAP) (Equation (2)). Microbial C limitation increases with vector length. Vector angles >45° represent microbial P limitation and vector angles <45° represent N limitation. Microbial P limitation increases with increasing vector angle and microbial N limitation increases with decreasing vector angle [26]. The equations are as follows:

$$vector\ length(\text{VL}) = \sqrt{(ln\ BG/ln[NAG + LAP])^2 + (ln\ BG/ln\ AP)^2} \qquad (1)$$

$$vector\ angle\ (\text{VA}) = Degrees(ATAN2(ln\ BG/ln\ AP, ln\ BG/ln[NAG + LAP])) \qquad (2)$$

### 2.6. Statistical Analysis

The mean and standard error of all data were calculated using SPSS Statistics 22 software. One-way analysis of variance and Duncan analysis were used to evaluate the differences in soil physical and chemical properties, extracellular enzyme activities and stoichiometry, and vector characteristics among different plots. Significance was evaluated at the level of alpha = 0.05. Figures were constructed using Origin Pro 2019.

Redundancy analysis (RDA) was performed using Canoco 5 software to illustrate the relationships between the soil and litter properties and extracellular enzyme activities and stoichiometry. Linear regression analysis was used to identify the factors among the soil physicochemical properties, litter C, N, and P, and soil extracellular enzyme activities that significantly explained the changes in vector characteristics.

## 3. Results

### 3.1. Soil and Litter Physiochemical Properties

Regarding the soil physicochemical properties (Figure 1), there were significant differences in soil moisture and nutrient content but not pH among the plots (Figure 1). Soil from the shrub and herb plots displayed higher moisture than that from the forest plots ($p < 0.05$). The levels of soil nutrients in the *Pinus* and shrub plots were higher than those in other plots, with higher values of $NH_4^+$-N, $NO_3^-$-N, and organic C in the former. However, the soil available P in the mixed-plantation plots was significantly higher than in the other plots ($p < 0.05$), indicating a higher availability of P. The litter contents of total C and total P were higher in the *Pinus* plots than in the other plots, indicating higher levels of litter nutrients in the *Pinus* plots.

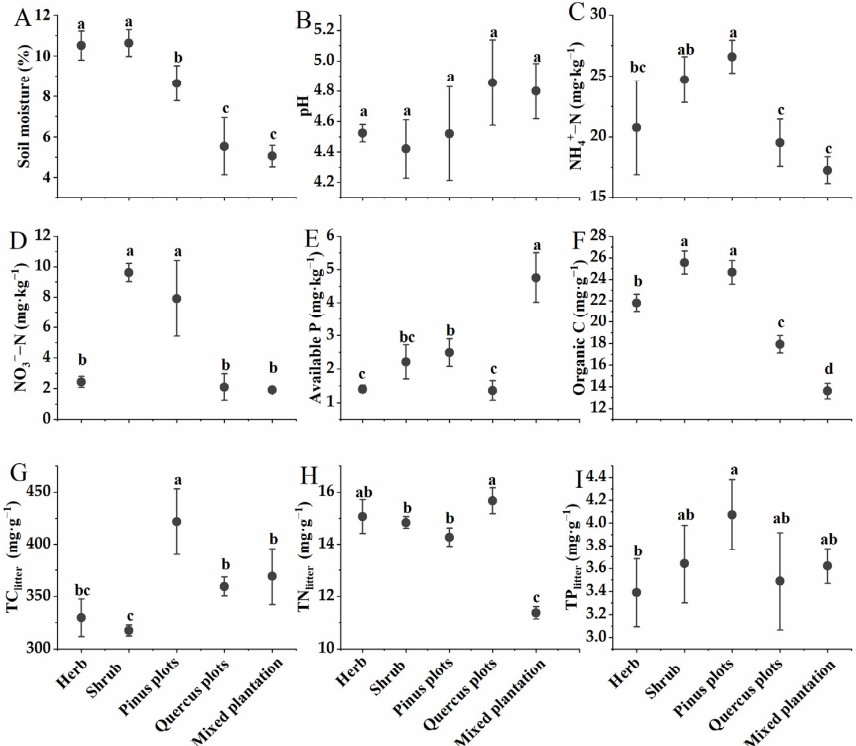

**Figure 1.** Descriptive statistics of soil and litter properties. $TC_{litter}$: litter total carbon, $TN_{litter}$: litter total nitrogen, $TP_{litter}$: litter total phosphorus. (**A–F**) Soil physical and chemical indicators. (**G–I**) Litter nutrition indicators. Values are the means ± standard error. The different letters indicate significant differences between treatments according to the Tukey test; lowercase letters (such as a, b, c, d) indicate significant differences ($p < 0.05$) among the five sampling plots.

## 3.2. Extracellular Enzyme Activities

The contents of extracellular enzymes with activities in C, N, and P acquisition differed among the different vegetation communities (Figure 2). The highest level of AP in soil was observed in the herb plots, while the level of the N-acquiring enzymes of LAP and NAG were lower in these plots than in the forest plots, indicating that the soil microorganisms in the herb plots had more P requirements and fewer N requirements than those in the forest plots. The highest BG and LAP activities in soil were observed in the *Quercus* plots, indicating that the soil microorganisms in these plots had more C and N requirements. NAG activity was higher in the *Pinus* plots than in the other plots, indicating that the *Pinus* plots required more N to maintain microbial nutrient balance.

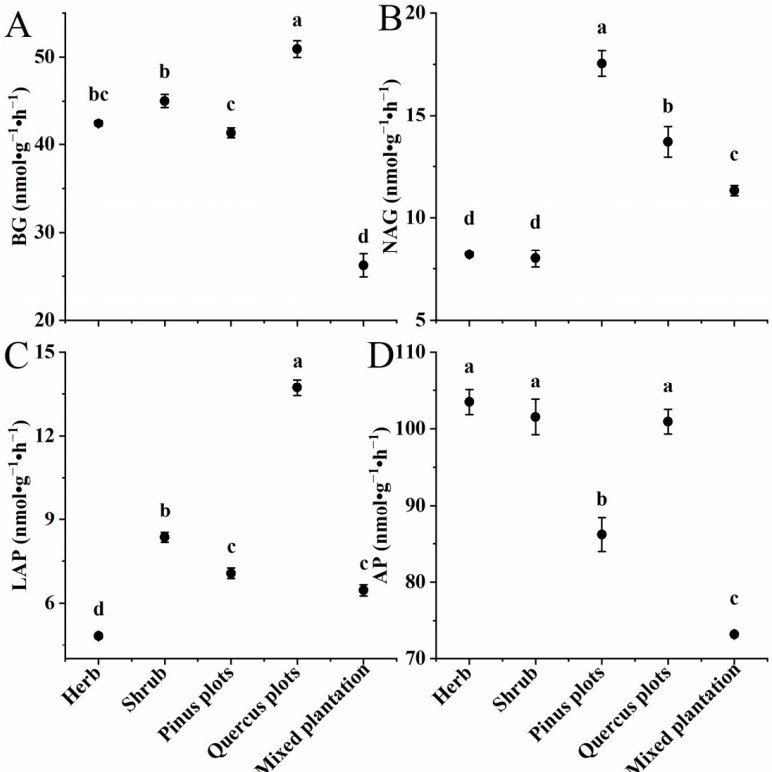

**Figure 2.** Changes in the extracellular enzyme activities. (**A**) The soil C-acquiring enzyme BG (β-1,-4-glucosidase), (**B**) The soil N-acquiring enzyme NAG(β-1,4-N-acetylglucosaminidase), (**C**) The soil N-acquiring enzyme LAP(L-leucine aminopeptidase), (**D**) The soil P-acquiring enzyme AP(alkaline phosphatase). Values are the means ± standard error. The different letters indicate significant differences between treatments according to the Tukey test; lowercase letters (such as a, b, c, d) indicate significant differences ($p < 0.05$) among the five sampling plots.

## 3.3. Vector Characteristics of Extracellular Enzyme Stoichiometry

The soil stoichiometric and vector characteristics of extracellular enzymes are shown in Figure 3. The soil extracellular enzyme stoichiometry scatter plot showed that all the data points fell within the C and P colimitation quadrant. All the lnBG/ln(NAG + LAP) values were above 1, all the values of lnBG/lnAP and ln(NAG + LAP)/lnAP were below 1, and all the vector angles were >45°, indicating that the soil microbial metabolism was limited by P; the microbial P limitation was greatest for the soil of the herb plots. Vector lengths (indicating microbial C limitation) were largest and smallest for the soils of the herb and mixed-plantation plots, respectively. Linear regression analysis revealed a significant positive correlation between microbial C limitation and microbial P limitation ($p < 0.01$). In summary, there were strong C and P limitations in the microbial communities in our study area. The C and P limitations in the herb plots were higher than those in the forest plots.

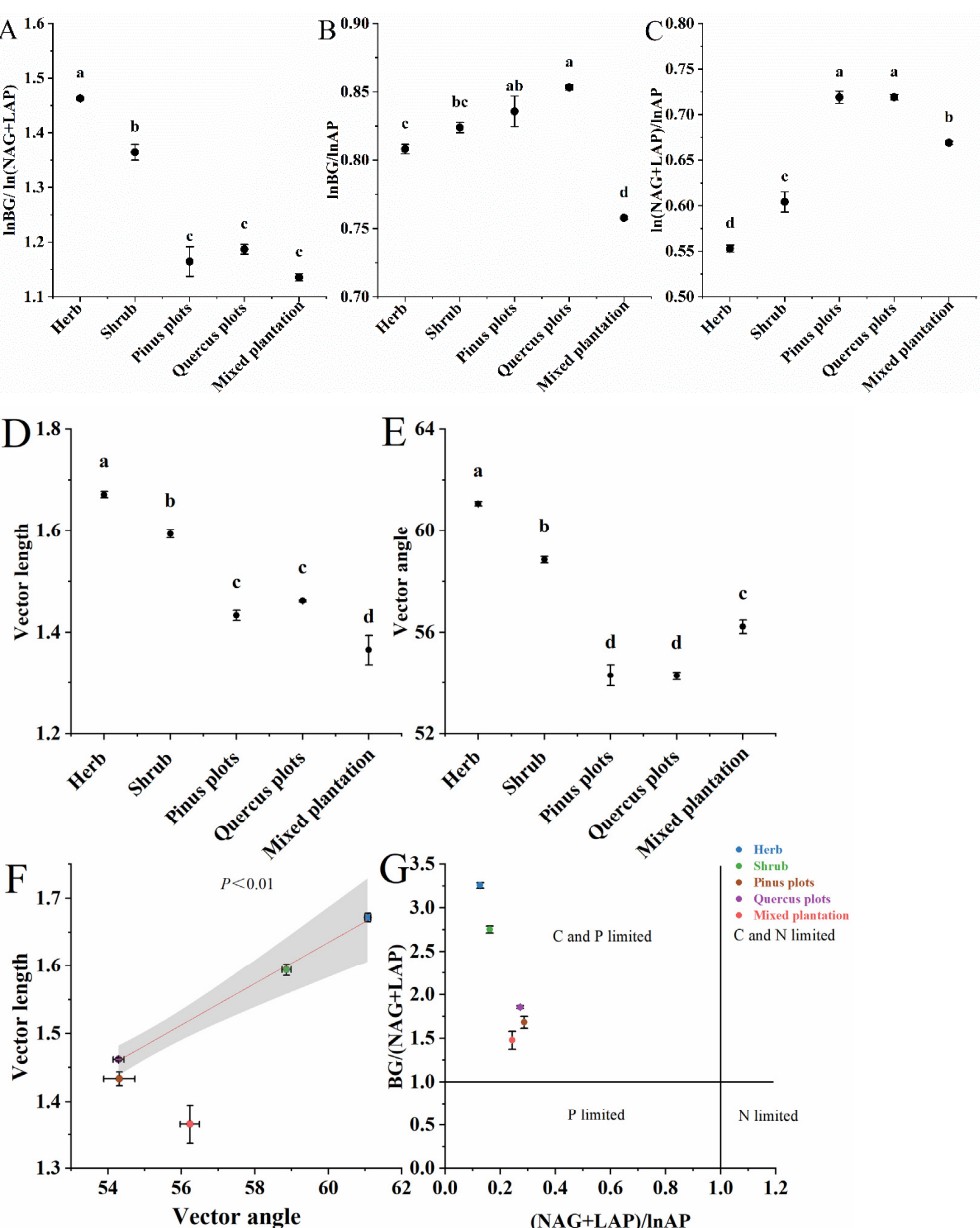

**Figure 3.** Extracellular enzyme stoichiometry of the relative proportions, (**A**–**E**) Changes in extracellular enzyme stoichiometry, vector length, and vector angle were calculated according to the ratios of the log-transformed BG, NAG + LAP, and AP. (**F**) Linear regression analysis to identify the relationships of microbial C limitation with microbial N or P limitation. Longer vector length indicates greater C limitation. A vector angle of <45° denotes N limitation; angles >45° denote P limitation. (**G**) Stoichiometric analysis of soil enzyme activities to identify potential P and N limitations in soil: by using 1 as a horizontal and vertical baseline along the axis of enzyme activity ratios ((NAG + LAP)/AP as the *x*-axis and BG/(NAG + LAP) as the *y*-axis), four quadrants representing microbial resource limitations (N limitation, P limitation, C and P limitations, and N and P limitations) were categorized. The different letters indicate significant differences between treatments according to the Tukey test; lowercase letters (such as a, b, c, d) indicate significant differences ($p < 0.05$) among the five sampling plots.

### 3.4. Relationships between the Limitation of Microbial Metabolism and Soil and Litter Properties

RDA was performed to evaluate the correlations among the soil physicochemical properties, litter nutrients, soil extracellular enzyme activities, and microbial metabolism limitations (Figure 4, Table 1). Soil and litter properties explained 93.53% of the variation in

extracellular enzymes in the RDA. The RDA revealed significant ($p < 0.05$) and positive relationships between $TN_{litter}$, $TC_{litter}$, and soil moisture and soil extracellular enzyme activities (Figure 4A and Table 1). Soil and litter properties explained 88.42% of the variation in extracellular enzyme stoichiometry in the RDA. The RDA revealed significant ($p < 0.05$) and positive relationships between $TC_{litter}$ and soil moisture and soil extracellular enzyme stoichiometry (Figure 4B and Table 1). Thus, the total C content of litter and soil moisture explained most of the variance in extracellular enzyme activities and their stoichiometry.

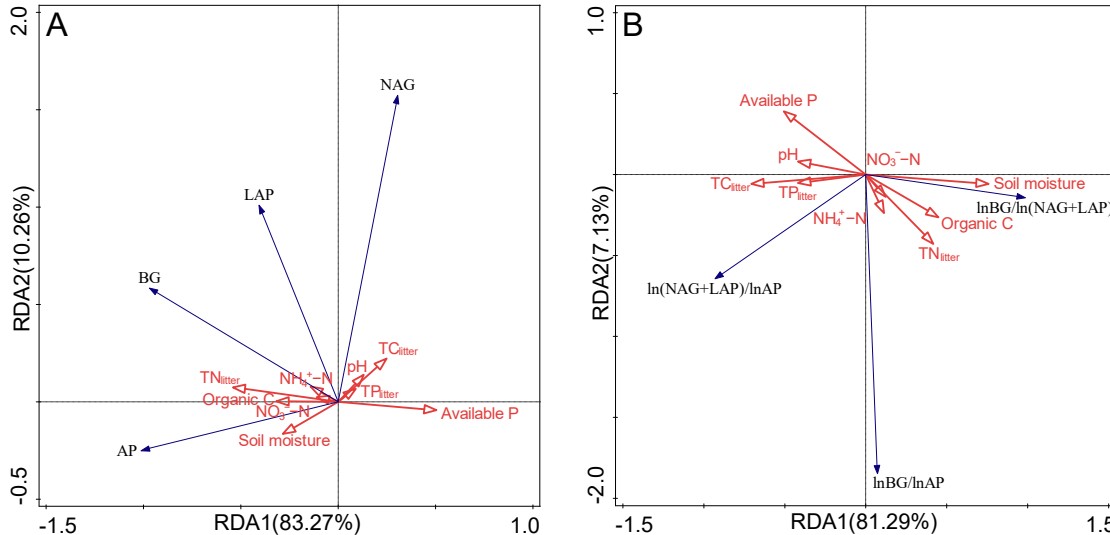

**Figure 4.** Redundancy analysis (RDA) used to identify the relationships between soil extracellular enzymes and soil and litter properties (**A**) and between soil extracellular enzyme stoichiometry and soil and litter properties (**B**).

**Table 1.** The explained rate of soil and litter properties to soil extracellular enzyme activities and the soil extracellular enzyme stoichiometry determined by RDA.

| Index | Soil Extracellular Enzyme Explains (%) | *p*-Value | Index | Soil Extracellular Enzyme Stoichiometry Explains (%) | *p*-Value |
|---|---|---|---|---|---|
| $TN_{litter}$ | 76 | 0.002 | Soil moisture | 52 | 0.006 |
| $TC_{litter}$ | 11.5 | 0.002 | $TC_{litter}$ | 22.4 | 0.002 |
| Soil moisture | 3.5 | 0.014 | Organic C | 6.1 | 0.070 |
| Organic C | 1.6 | 0.108 | $TN_{litter}$ | 4.4 | 0.094 |
| pH | 1.1 | 0.212 | $TP_{litter}$ | 1.3 | 0.376 |
| $TP_{litter}$ | 0.6 | 0.46 | $NH_4^+-N$ | 1.4 | 0.392 |
| Available P | 0.5 | 0.592 | $NO_3^--N$ | 0.6 | 0.636 |
| $NO_3^--N$ | 0.4 | 0.668 | Available P | 0.4 | 0.728 |
| $NH_4^+-N$ | 0.1 | 0.946 | pH | <0.1 | 0.996 |

The linear regression analysis showed that vector angles were positively correlated with soil moisture values ($p < 0.01$), which were significantly negatively correlated with the LAP ($p < 0.05$), NAG ($p < 0.001$), and the total C of litter, respectively ($p < 0.01$) (Figure 5). The linear regression analysis showed that vector lengths were positively correlated with soil moisture ($p < 0.01$), soil organic C ($p < 0.05$), AP ($p < 0.001$), and the total N of litter, respectively ($p < 0.05$), and that they were significantly negatively correlated with the available P ($p < 0.001$), NAG ($p < 0.01$), and the total C of litter, respectively ($p < 0.05$) (Figure 6, Table S2).

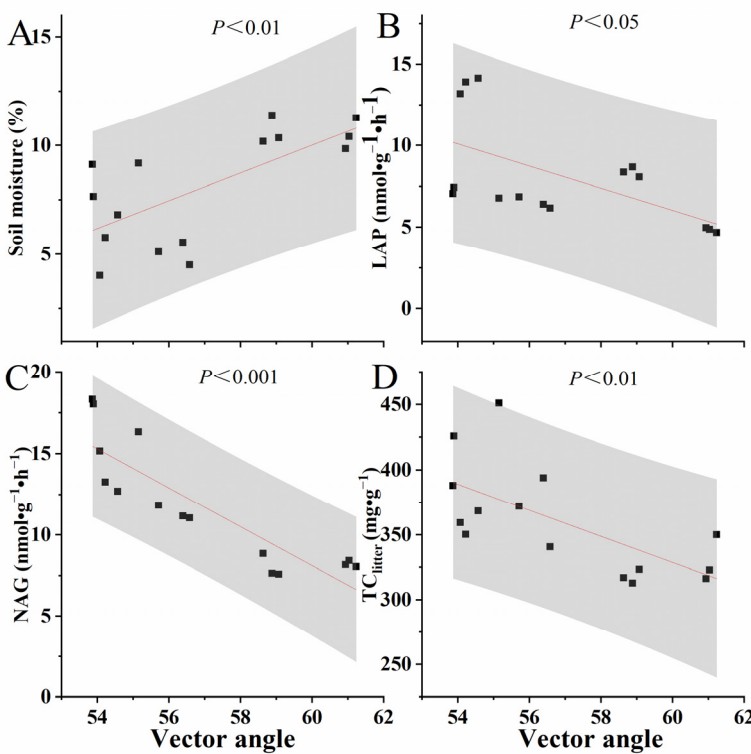

**Figure 5.** Linear regression analysis between the soil physical properties (**A**), soil extracellular enzymes (**B**,**C**), litter nutrients, (**D**) and vector angles. Red lines indicate the model fits between the vector angles and the properties; grey areas represent the 95% confidence intervals of the models.

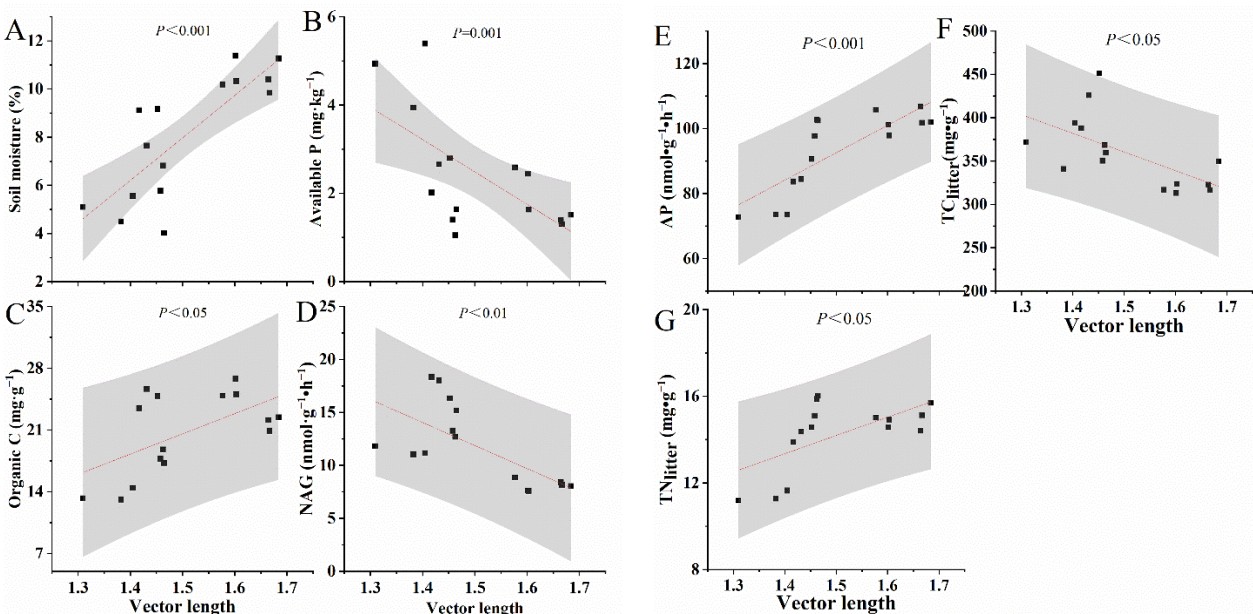

**Figure 6.** Linear regression analysis between the soil physical properties (**A**), soil nutrients (**B**,**C**), soil extracellular enzymes (**D**,**E**), litter nutrients, (**F**,**G**) and vector lengths. Red lines indicate the model fits between the vector lengths and the properties; grey areas represent the 95% confidence intervals of the models.

## 4. Discussion

### 4.1. Relative Limitations of Microbial Metabolism

This study determined the relative soil microbial metabolism C and P limitations in the Yimeng Mountain area, China (Figure 3F,G), and hypothesis 1 was supported. This

finding is supported by two lines of evidence. Soil extracellular enzyme stoichiometry reflects soil microbial nutrient requirements and limitations [29,30]. In this study, the stoichiometric ratio of C, N, and P of soil enzymes after logarithmic transformation was 1:0.80:1.14, which deviates from the global enzymes ratio of 1:1:1 [7]. lnBG/lnAP and ln(NAG + LAP)/lnAP were both lower than 1, and the vector angles were also greater than 45° (Figure 3B–D). This result indicates that the soil microorganisms have a high demand for P convertase activity in this area, and P was relatively limited, which was consistent with the research results of Cui which found a grassland ecosystem to be limited by microorganism P [16,23]. The soil extracellular stoichiometry scatter plot provided more intuitive evidence that the soil microbes under all plots were colimited by C and P. This result is consistent with the findings of Chen [9] and Bai [31] on C and P limitation in grassland and forest ecosystems. Differences in vegetation types and soil characteristics can also all create this limitation [10]. Under acidic soil conditions, P is not easily absorbed and utilized by plants and microorganisms [32], and the vector length is positively correlated with the vector angle of the plots, indicating that soil microorganisms' C and P were limiting factors for plant growth in the Yimeng Mountain area, China.

The vector angles of herb and shrub plots were 61.07° and 58.86°, respectively, which were significantly higher than the 54.29°, 54.31°, and 56.23° angles of the *Quercus acutissima*, *Pinus thunbergii* and mixed-plantation plots (Figure 3D), indicating that the P limitations of herb and shrub plots were higher than those of forest plots, and hypothesis 2 was rejected. This result was different from those of Huang's [24] research on forest succession. Our results suggested that the herb to forest stage C and P limitations gradually decreased, which may be caused by the domination of gramineous herb plots investing more AP to access their own required nutrients [33], indicating that the P content was relatively lower (Figure 1E), and the microbial P limitation was relatively higher. The P limitation of the mixed plantation was higher than that of the pure forest, and the C limitation was not as strong as that of the pure forest. The artificial forest in the experimental area had a short afforestation time and different soil conditions, being more arid and barren (Figure 1). Different from the limitation of microbial C and P, the mixed-plantation plots may change from P limitation to N limitation. Another consideration is that the soil enzymes lnBG/ln(NAG + LAP) showed levels lower than the global average of 1.41 except for herb plots; lnBG/lnAP and ln(NAG + LAP)/lnAP values were both higher than the global average (0.62 and 0.44) (Figure 3 A–C) [26]. In this study, the vector length of herb plots increased compared with that of forest plots, and both lnBG/lnAP and ln(NAG + LAP)/lnAP decreased. According to resource allocation theory [26], when there is a certain element restriction, microorganisms will weigh their own needs and invest more resources into obtaining the limited elements. This showed that the investment of herbaceous community microorganisms in obtaining enzymes for N was relatively less than that for obtaining enzymes for C, microorganisms will put more resources into C acquisition enzymes [34], and the growth of microorganisms may be relative by C.

### 4.2. Soil Physical and Chemical Properties and Litter Nutrition Affects Microbial C and P Limitations

RDA and linear regression analysis indicated that the dynamics of soil extracellular enzyme activities were explained by the properties of the soil and litter. Soil moisture and litter C and N explained most of the variation in soil extracellular enzyme activities (Figure 4), which is consistent with previous studies [35]. Soil moisture and litter total C explained most of the variation in soil enzyme stoichiometry, whereas soil organic C and available P predominantly controlled vector characteristics. There was a significant correlation between soil extracellular enzyme activities and stoichiometry and vector characteristics. Therefore, the influencing factors on soil microbial metabolism limitations can be explained by soil physical and chemical properties and litter total C and N, which supports our third hypothesis.

In this study, soil moisture was found to be positively correlated with enzyme activities and vector characteristics (Figures 4A and 5). Soil moisture controls the distribution and transmission of materials and energy within and outside the ecosystem and plays an important role in the maintenance of ecosystem productivity and ecosystem services [36]. There was a positive correlation between soil moisture and the P acquisition enzyme AP, and soil moisture determines the diffusion rate of enzymes, substrates, and products [37], which is a key factor in microbial metabolism [38]. When soil moisture is low, enzyme activity is relatively low, and a long-term lack of water will also reduce the enzyme activity and inhibit microbial metabolism [39]. The range of soil moisture in this study was 5.04–10.63% (Figure 1A), which indicates a moderately dry period of soil [40], which will affect soil water availability and increase ecological vulnerability [41]. This was also an important reason for microbial metabolism limitations.

Soil chemical properties have been identified as key factors influencing microbial metabolism. In this study, AP, BG, and vector length were negatively correlated with soil available P and positively correlated with soil organic C (Figure S2 and Figure 5), suggesting different effects of soil C and P on microbial nutrient limitations. The results of linear regression analysis further indicated that soil available nutrient contents influence microbial C limitation. Additionally, the high sand content in the sample plots (Table S1) was significantly negatively correlated with soil moisture and positively correlated with soil available P (Figure S2), indicating that high sand content affects soil water storage performance and nutrient sequestration [42]. A global meta-analysis showed that soil organic C content is affected by vegetation type in arid and semiarid regions [43], and the average soil organic C contents of mixed-plantation and herb plots were lower than those of other plots. Thus, with different accumulation rates of soil organic C [43] and with different properties in litter and soil [44], under the influence of the soil aggregate structure [45], the C source provided was reduced, which significantly influenced soil C sequestration and nutrient cycling. Changes in the availability of soil nutrients influence extracellular enzyme activities and stoichiometry [12,46], thereby influencing the limitations of microbial metabolism. Another consideration is that microbial activities can provide nutrients for plants, but when nutrients contents are low, microbes will also compete with plants for nutrients [38]. Therefore, the availability of soil nutrients is another important reason for microbial nutrient limitations [47]. Therefore, soil physical and chemical properties can be used to explain the variation in soil extracellular enzyme activities and microbial nutrition limitations.

This study found that litter total C and N had a significant correlation with soil extracellular enzyme activities, especially those of AP and BG. Research on four soil extracellular enzyme activities (Figure 2) found that enzymes related to C, N, and P were different in the communities. As a result of the different soil properties of each plot (Figure 1), the litter decomposition production of saprotrophic basidiomycetes and archaea affected microbial growth and extracellular enzyme activities [19,48–50]. Furthermore, there was a correlation between extracellular enzymes (Figure S1). The different enzymes have similar roles in soil fertility, and soil extracellular enzymes jointly promote the degradation of lignin, the formation of humus, and the release of litter-like nutrients [19,51]. Therefore, litter nutrients can be used to explain variation in soil extracellular enzyme activities. Therefore, soil moisture, soil organic C, soil available P, and litter total C and N can indirectly affect extracellular enzyme stoichiomety and vector characteristics by affecting extracellular enzyme activities [23], thus affecting the nutritional limitations of microbial metabolism.

## 5. Conclusions

In summary, the stoichiometric ratio of C, N, and P after logarithmic transformation of soil extracellular enzymes was 1:0.80:1.14, which deviates from the global ecological ratio of 1:1:1. The lnBG/ln(NAG + LAP) was higher than 1 and the lnBG/lnAP and ln(NAG + LAP)/lnAP were both lower than 1, which indicated that microbial metabolism tended to invest more resources in C- and P-acquisition enzymes. The limitations were directly mediated by soil moisture and litter total C and N and soil available nutrients.

The vector angles and vector lengths of herb and shrub plots were relatively high, which also indicated that the resource inputs of herb and shrub plots under microbial limitations were higher than in forest plots and that the microbial C and P limitations were stronger, which was mainly attributed to changes in soil moisture and soil organic C and available P. Therefore, in this study, soil physical and chemical properties and litter total C and N were used to assess microbial C and P limitations. Our results provide useful insights into understanding the characteristics of microbial metabolism in arid and barren mountain forest ecosystems and improve our understanding of the forest ecosystem C cycle.

**Supplementary Materials:** The following are available online at https://www.mdpi.com/article/10.3390/f13050692/s1, Figure S1: Location of the state-owned Dawa Forest Farm research station in the Yimeng Mountain area of Linyi city in China, Figure S2: Correlation heat map of soil physicochemical properties, litter total C, N, P, extracellular enzyme, extracellular enzyme stoichiometry, and microbial nutrient limitations, Table S1: Distribution characteristics of soil particle size in different plots, Table S2: Linear regression analysis of the relationship between microbial metabolism limitation, soil, and litter properties.

**Author Contributions:** L.W., K.L., J.G. (Jianyao Guo), M.L. and C.L. conceived and designed the study. L.W., X.L., J.G. (Jinhui Gao), L.M. and J.W. performed the experiments. L.W., K.L., J.G. (Jianyao Guo), X.L., J.G. (Jinhui Gao), C.L. and M.L. analyzed the data. L.W., K.L., J.G. (Jianyao Guo), X.L., J.G. (Jinhui Gao), M.L., J.W., M.L. and C.L. wrote the manuscript. L.W. and K.L. share first authorship. All authors have read and agreed to the published version of the manuscript.

**Funding:** This work was supported by the Forestry Science and Technology Innovation Project of Shandong Province (no. 2019 LY005); National Natural Science Foundation of China (no. 31570705); Evaluation and Promotion Project of Forest Resource Carbon Sink Capacity under Carbon Neutralization Background of Shandong Forestry Protection and Development Service Center.

**Institutional Review Board Statement:** Not applicable.

**Informed Consent Statement:** Not applicable.

**Data Availability Statement:** The data presented in this study are available on request from the first author.

**Conflicts of Interest:** The authors declare no conflict.

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
