# Peer review of "Extracellular Enzyme Stoichiometry Reveals Soil Microbial Carbon and Phosphorus Limitations in the Yimeng Mountain Area, China"

_forests, doi:10.3390/f13050692_

Round 1
Reviewer 1 Report
Forests-1565561: "Ecoenzymatic stoichiometry reveals soil microbial C and P limitation in the Yimeng Mountain area, China".
The subject of the manuscript is consistent with the scope of the Journal. Manuscript present many interesting results about an important subject. The present paper is prepared in the usual manner for scientific work, both the division into chapters collected results in the form of tables and figures. The authors applied correct analytical methods and received many interesting results. The obtained results do not raise any substantive objections.
Author Response
Responses to Reviewers 1 Comments
MS Type: Article
Title: Extracellular enzyme stoichiometry reveals soil microbial carbon and phosphorus limitation in the Yimeng Mountain area, China
Dear Revision Quality Check:
Thank you very much for your attention and the referees’ evaluation and comments on our paper. These comments were very valuable and helpful for revising and improving our article, as well as guiding the significance of our research. We have studied the comments carefully and have made corrections in the manuscript.
We tried our best to improve the manuscript and made revised in the manuscript. These changes did not influence the content or framework of the paper. Here, we did not list the changes but marked them in red in our revised paper.
We appreciate the hard work of the editor and reviewers and hope that the corrections will meet with your approval.
Once again, thank you very much for your comments and suggestions.

Reviewer 2 Report
Dear Authors, kindly check English sentence structure, Grammar errors, throughout the manuscript and suggested modified changes as per the comments given below:
- Line 12, the author must write the “C” in full form in the first sentence.
- Line 15, what are the litter characteristics?
- Line12-30, the author should write a short methodology for the study.
- Line 18, What is the full form of “EEA” and try to write initially full form for all abbreviations used in the manuscript.
- Line 28-30, the Author should revise the statement as a future recommendation.
- Line 37, the word ‘’Therefore’’ should revise to “However”.
- Line 89-90, the statement regarding thickness and moisture needs to clearer, how thickness affects the soil properties and Moisture changes time by time, so the statement should be clear.
- Line 99-100, The aim of the study is not clear, so it should be in with the specific objective of the study.
- Line 101-106 corrected the spelling of Hypotheses as “Hypothesis” and write the number as Ho, H1, and so on.
- Line 155-116, the Author suggested incorporating some supporting references regarding the soil properties of the study area.
- Line 142, Soil Available Phosphorus should write as (AP) it is the scientific abbreviation.
- Line 183, the author suggested to mentioned, which R package was used for analyzing the data.
- Figures 1, 2, 3, 5 must be replaced with higher resolution and increased the size of fonts for the XY axis in the charts/graph.
- Check the sentence in the whole section of the introduction and discussion for further improvement in the revised
- Line 418-431, the overall conclusion ok but should be clear the hypothesis is supported or not and suggested writing a statement on the planning of future study or recommendations to policymaker and research community.
Best of Luck

Author Response
Responses to Reviewers 2 Comments
MS Type: Article
Title: Extracellular enzyme stoichiometry reveals soil microbial carbon and phosphorus limitation in the Yimeng Mountain area, China
Dear Revision Quality Check:
Thank you very much for your attention and the referees’ evaluation and comments on our paper. These comments were very valuable and helpful for revising and improving our article, as well as guiding the significance of our research. We have studied the comments carefully and have made corrections in the manuscript, and the responses to the reviewers’ comments are as follows.
- Line 12, the author must write the “C” in full form in the first sentence.
Responses: We have revised the “carbon” in full form.
- Line 15, what are the litter characteristics?
Responses:The litter characteristics represents the C, N, P of the litter, which has been modified to “the total carbon, nitrogen and phosphorus content of litter”.
- Line12-30, the author should write a short methodology for the study.
Responses: We have revised the article as required. A methodology for the study that are “The total carbon(C), nitrogen(N) and phosphorus(P) content of litter, soil physical and chemical properties were analyzed. The activities of four extracellular enzymes β-1,4-glucosidase (BG), β-1,4-N-acetylglucosaminidase (NAG), L-leucine aminopeptidase (LAP) and acid phosphatase (AP) related to soil C, N and P cycle were analyzed. Models of extracellular enzymatic stoichiometry was used to study and compare the metabolic limitation of soil microorganisms in different plots, and the driving factors of microbial metabolic limitation were explained”.
- Line 18, What is the full form of “EEA” and try to write initially full form for all abbreviations used in the manuscript
Responses: “EEA” stands for soil extracellular enzyme activity. We have revised EEAC:N to ln(BG)/ ln(NAG + LAP).
- Line 28-30, the Author should revise the statement as a future recommendation
Responses: The statement as a future recommendation have been re-provided the after results section of abstract in the manuscript.
- Line 37, the word ‘’Therefore’’ should revise to “However”.
Responses: We have revised “Therefore” to “However”.
- Line 89-90, the statement regarding thickness and moisture needs to clearer, how thickness affects the soil properties and Moisture changes time by time, so the statement should be clear.
Responses: We have revised to “At present, the research on the soil in Yimeng mountain area mainly focuses on the spatial distribution characteristics of soil thickness and the analysis of soil hydrological characteristics on the function of soil and water conservation.”
- Line 99-100, The aim of the study is not clear, so it should be in with the specific objective of the study.
Responses: We have clarified the aim of the study.
- Line 101-106 corrected the spelling of Hypotheses as “Hypothesis” and write the number as Ho, H1, and so on.
Responses: We have revised spelling, and have revised the number as H1, H2, H3.
- Line 155-116, the Author suggested incorporating some supporting references regarding the soil properties of the study area.
Responses: The soil thickness is the average of the field surveys in this area.
- Line 142, Soil Available Phosphorus should write as (AP) it is the scientific abbreviation.
Responses: In order to distinguish acid phosphatase (AP), SAP has been changed to soil available phosphorus in full form.
- Line 183, the author suggested to mentioned, which R package was used for analyzing the data.
Responses: We have revised the statistical analysis section.
- Figures 1, 2, 3, 5 must be replaced with higher resolution and increased the size of fonts for the XY axis in the charts/graph.
Responses: We have redrawn with higher resolution and increased the size of fonts for the XY axis in the charts/graph.
- Check the sentence in the whole section of the introduction and discussion for further improvement in the revised.
Responses: We have checked the sentence in the whole section of the introduction and discussion and have made revisions.
- Line 418-431, the overall conclusion ok but should be clear the hypothesis is supported or not and suggested writing a statement on the planning of future study or recommendations to policymaker and research community.
Responses: We have revised the entire conclusion section.
We tried our best to improve the manuscript and made revised in the manuscript. These changes did not influence the content or framework of the paper. Here, we did not list the changes but marked them in red in our revised paper.
We appreciate the hard work of the editor and reviewers and hope that the corrections will meet with your approval.
Once again, thank you very much for your comments and suggestions.

Reviewer 3 Report
In general, the research of this manuscript is very interesting and in line with the scope of the special issue “Forest biodiversity and Ecosystem Stability”.
Abstract
The ‘Abstract’ was abruptly concluded. I suggest including possible implications of your study. There is a lack of specific data. It is suggested to express it with data.
Line 20: Why the community soil microorganisms were relatively limited by C, compared with which community or other research?
Line 21-26: I think the expression here is not clear. What does the EEA refers to, the first occurrence in the abstract should be the full name plus abbreviation. Hypothetical responses and differences between communities are not reflected in the abstract. In general, the statement of results and conclusions should be more specific.
Line 27- 28: Does the “nutrient cycling and C sequestration” refers to soil? Please specify.
Introduction
The ‘Introduction’ was well conducted, although I felt authors should restrict it to the essential aspects that are needed to support the target answers and hypothesis. It is essential that transitions between subsequent paragraphs should be perfect.
In this part, “ecological enzymes”, “ecological enzymes activity” and “soil enzymes” should be unified.
Line 64-65: “C:P ratio” is soil C:P or EEAC:P?
Line 70-71: The sentence should be rewritten, it is incomplete.
Line 76: C:P and N:P are soil or EEA? I think these should be marked clear for litter, soil, and enzymes.
Line 105-106:If the Hypothesis 3 should be affected by soil physical and chemical properties and litter nutrients? I think the statement for hypothesis 3 is too specific.
Table 1
Does the Shannon Wiener index refers to arbor, shrub, herb or the all plant? I think it can be analyzed with soil, litter and enzymes indexes to explore the relationships.
Materials and methods
Line 129-132: Please specify the sampling number of shrub and herb.
Line 169: How long does the vector length refers to microbial C limitation? How to define it?
Line 173-177: Each variable in the formula should be explained below the formulas and written in italics.
Line 194: The preposition “with” should be changed to “of” or others.
Results
The results only described which community was low or high, lacking in-depth analysis and summary.
Figure 1: Does the black spot refers to the mean value? The shape of the figure should be explained below the figures.
Figure 3: I think the one-way Anova analysis and multiple comparisons for vector angle and vector length should be supplemented.
Figure 4: In the figure, there are two EEAC:N, please check it. I think RDA analysis can give the results of each variables explanation and significance, these can be stated in the results.
Line 269-278: This part should be rewritten, I think the author should analysis the relationship between microbial metabolism variables(blue) and soil and litter characteristics(red), but not the relationship between soil and litter characteristics.
Figure 5: Stepwise regression can include main influential variables into the model, but the graph in this paper is only a linear regression between a variable and the dependent variable. I think BG, AP and Vector Length may be jointly influenced by multiple variables. If it is not clear, SEM can be tried to carry out to specific the direct and indirect effects between variables.
Discussion
The part needs to be in-depth discussion.
The part of 4.1: If the soil physical and chemical properties are related to the community species biodiversity or the key species? In addition, related research about soil physical and chemical properties and litter nutrient should be supplemented to compare and discuss with the results.
Line297-298: It is obscure, what does the author want to express?
L299: Is the thesis to be discussed? The presentation needs to be improved.
Line30-302: Hu’s research studied the sub alpine forests. Is it comparable with this study? Is it in the same latitude or region?
Line305: I think the high ammonia and nitrate levels in Pinus Thunbergii forest may be related to its existence of mycorrhizal fungi. And if is it related to the understory herbaceous diversity? If a figure composed of multiple little figures, please add “a, b, c...” in the little figures and indicate them in the corresponding description of the results and discussion.
Line307: Does the “N content” and “Herb P” for soil or what, TN and TP? Please specify.
Line 308-313 Do these sentences can explain the previous statement? This research did not involve about aggregate structure?
Line 315-331: This paragraph lacks comparison with other studies and in-depth discussion.

Author Response
Responses to Reviewers 3 Comments
MS Type: Article
Title: Extracellular enzyme stoichiometry reveals soil microbial carbon and phosphorus limitations in the Yimeng Mountain area, China
Dear Revision Quality Check:
Thank you very much for your attention and the referees’ evaluation and comments on our paper. These comments were very valuable and helpful for revising and improving our article, as well as guiding the significance of our research. We have studied the comments carefully and have made corrections in the manuscript, and the responses to the reviewers’ comments are as follows.
- The ‘Abstract’ was abruptly concluded. I suggest including possible implications of your study. There is a lack of specific data. It is suggested to express it with data.
Responses: The " Abstract " has been revised. We have added the research's possible impact and specific data.
- Line 20: Why the community soil microorganisms were relatively limited by C, compared with which community or other research?
Responses: Take Hill’s (2012) and Zheng’s (2020) research as a reference. “Microbial resource limitation was judged based on the scatter plot between (LAP+NAG)/AP and BG/(LAP+NAG) with the Four parts in the plot representing soil microbial N limitation, P limitation, C and N limitation, C and P limitation, respectively. This requires plotting the ratio of (LAP+NAG)/AP on the x-axis and BG/(LAP+NAG) ratio on the y-axis and denoting horizontal and vertical coordinates of 1 as baselines.”
References
Hill, A.; Brian H.; Seifert, L.R.; May, A.A.; Tarquinio, E. Microbial enzyme stoichiometry and nutrient limitation in US streams and rivers. Ecol. Indic. 2012, 18, 540-551.
Zheng, L.; Chen, H.; Wang, Y.Q.; Mao, Q.G.; Zheng, M.H.; Su, Y.R.; Xiao, K.C.; Wang, K.L.; Li, D.J. Responses of soil microbial resource limitation to multiple fertilization strategies. Soil Till. Res. 2020, 196, 104474.
- Line 21-26: I think the expression here is not clear. What does the EEA refers to, the first occurrence in the abstract should be the full name plus abbreviation. Hypothetical responses and differences between communities are not reflected in the abstract. In general, the statement of results and conclusions should be more specific.
Responses: We have revised abbreviation of the first occurrence to use full name plus abbreviation in the abstract. The result and conclusions have revised in the abstract.
- Line 27- 28: Does the “nutrient cycling and C sequestration” refers to soil? Please specify.
Responses: The “nutrient cycling and C sequestration” refers to soil, we have revised to “soil nutrient cycling”
- In this part, “ecological enzymes”, “ecological enzymes activity” and “soil enzymes” should be unified.
Responses: We have uniformly revised “ecological enzymes”, “ecological enzymes activity” and “soil enzymes” as “soil extracellular enzyme”.
- Line 64-65: “C:P ratio” is soilC:P or EEAC:P?
Responses: It was C:P of extracellular enzyme. We have revised.
- Line 70-71: The sentence should be rewritten, it is incomplete.
Responses: We have revised to “Vector characterization of extracellular enzymes to evaluate the nutritional requirements of microbes and clarify the relative restriction of microbial metabolism.”
- Line 76: C:P and N:P are soil or EEA? I think these should be marked clear for litter, soil, and enzymes.
Responses: The C:P and N:P were soil and we have revised.
- Line 105-106:If the Hypothesis 3 should be affected by soil physical and chemical properties and litter nutrients? I think the statement for hypothesis 3 is too specific.
Responses: The hypothesis has been revised to “The soil microbial limitations of the communities were affected by changes in soil physical and chemical properties and litter nutrients.”
- Table 1 Does the Shannon Wiener index refers to arbor, shrub, herb or the all plant? I think it can be analyzed with soil, litter and enzymes indexes to explore the relationships.
Responses: The Shannon Wiener index refers to all plant in plot.
- Line 129-132: Please specify the sampling number of shrub and herb.
Responses: The sampling number of shrub and herb the same as arbor, we have supplemented the soil sampling of shrub and herb.
- Line 169: How long does the vector length refers to microbial C limitation? How to define it?
Responses: Take Zheng’s (2020) and Chen’s (2019) research as a reference. BG/(LAP+NAG)>1 and (LAP+NAG)/AP<1.
- Line 173-177: Each variable in the formula should be explained below the formulas and written in italics.
Responses: Each variable in the formula have explained and written in italics.
- Line 194: The preposition “with” should be changed to “of” or others.
Responses: We have changed “with” to “of”.
- The results only described which community was low or high, lacking in-depth analysis and summary.
Responses: We have re-analyzed and summarized in-depth of results.
- Figure 1: Does the black spot refers to the mean value? The shape of the figure should be explained below the figures.
Responses: Figure has been redrawn and the shape of the figure has explained below the figures.
- Figure 3: I think the one-way Anova analysis and multiple comparisons for vector angle and vector length should be supplemented.
Responses: We have supplemented the the one-way Anova analysis and multiple comparisons for vector angle and vector length.
- Figure 4: In the figure, there are two EEAC:N, please check it. I think RDA analysis can give the results of each variables explanation and significance, these can be stated in the results.
Responses: Figure has been redrawn, and we have supplemented the table of results about each variables explanation and significance.
- Line 269-278: This part should be rewritten, I think the author should analysis the relationship between microbial metabolism variables(blue) and soil and litter characteristics(red), but not the relationship between soil and litter characteristics.
Responses: “3.4.The relationship between the limitation of microbial metabolism and the properties of soil and litter”, this part has re-written and analyzed.
- Figure 5: Stepwise regression can include main influential variables into the model, but the graph in this paper is only a linear regression between a variable and the dependent variable. I think BG, AP and Vector Length may be jointly influenced by multiple variables. If it is not clear, SEM can be tried to carry out to specific the direct and indirect effects between variables.
Responses: This part has redrawn and analyzed, and supplemented the correlation for all indicators.
- The part needs to be in-depth discussion.
Responses: All discussion sections have been re-analyzed in-depth.
- The part of 4.1: If the soil physical and chemical properties are related to the community species biodiversity or the key species? In addition, related research about soil physical and chemical properties and litter nutrient should be supplemented to compare and discuss with the results.
Responses: We have re-analyzed in 4.2 of discussion section.
- Line297-298: It is obscure, what does the author want to express?
Responses: In order to account for the differences of available nutrients content in different plots, it has been re-analyzed in 4.2 of the discussion section.
- L299: Is the thesis to be discussed? The presentation needs to be improved.
Responses: The discussion has been re-analyzed.
- Line30-302: Hu’s research studied the sub alpine forests. Is it comparable with this study? Is it in the same latitude or region?
Responses: We have revised and re-analyzed.
- Line305: I think the high ammonia and nitrate levels in Pinus Thunbergii forest may be related to its existence of mycorrhizal fungi. And if is it related to the understory herbaceous diversity? If a figure composed of multiple little figures, please add “a, b, c...” in the little figures and indicate them in the corresponding description of the results and discussion.
Responses: We have re-analyzed and add “A, B, C. in the little figures and indicate them in the corresponding description of the results and discussion.
- Line307: Does the “N content” and “Herb P” for soil or what, TN and TP? Please specify.
Responses: The “N content” is litter total N content of mixed plantation, “Herb P” is litter total P of herb.
- Line 308-313 Do these sentences can explain the previous statement? This research did not involve about aggregate structure?
Responses: We have added the characteristics of soil particle size in different plots in supplementary materials.
- Line 315-331: This paragraph lacks comparison with other studies and in-depth discussion.
Responses: We have revised this paragraph and in-depth discussion in 4.2 of the discussion section.
We tried our best to improve the manuscript and made revised in the manuscript. These changes did not influence the content or framework of the paper. Here, we did not list the changes but marked them in red in our revised paper.
We appreciate the hard work of the editor and reviewers and hope that the corrections will meet with your approval.
Once again, thank you very much for your comments and suggestions.

Reviewer 4 Report
Ecoenzymatic stoichiometry reveals soil microbial C and P limitation in the Yimeng Mountain area, China
Wang et al.
In this manuscript the authors reported the findings and analysis of potential soil extracellular enzyme activities under various vegetation types and correlated with soil physico-chemical properties. The topic is certainly interesting as both vegetation and soil properties were considered to explain enzymatic stoichiometry. The article is a descriptive one and based on only potential enzyme activities, as a proxy for microbial metabolic limitations without microbial community and explain it as microbial metabolic in however, I think that methodological novelty and in-depth analysis of enzymatic stoichiometry will be of interest to the readers of forests. Although the authors followed a perfect design to investigate how the vegetation succession and soil properties that shape up the soil extracellular enzymes, there are many correctable weaknesses in the manuscript. Overall, the results and discussions section need to rearrange and the whole documents need to check for English language errors. The manuscript needs major correction.
Introduction:
- Line 41-43: “The level…. but also characterize …..but its role is also…” Please split the sentence
- Line 45: “…organic carbon also acts on a variety of microorganisms to promote enzyme activity” the sentence doesn’t mean any strong sense.
- In entire manuscript the word “ecological” “eco- enzymatic” excessively and sometime unnecessarily, for example: “soil ecological enzyme activity (Line 48)”, Line 50-52: “Ecoenzymatic stoichiometry (is it differ if we say Soil enzymatic stoichiometry?) theory combines ecological metabolism theory and ecological stoichiometry theory to estimate the limitations of ecosystem microbial metabolism from the perspective of enzyme activity”. I think the whole manuscript needs to revise for such excess words.
- Line-66: “Therefore……. different environmental conditions is very …. Ecosystems”. Irrelevant because environmental condition was not considered here.
- Line 72-73: Need reference.
- Line 96: “In this study, the soil of five communities was studied” What does mean by “soil of five communities”? Sentence structure was not correct, plural noun and verb and repeated use of study.
- Line 96-101: should be transfer to Materials & methods section.
Materials & methods
- Line 110: “The mean annual temperature of this region is 12.8 °C, the mean annual precipitation is 600 mm, and the area has a warm temperate continental monsoon climate and a frost-free period of 191 d.” Please make it short avoid repeated words!
- Line 114 &118: “arbor soil” is not generally used, better is “forest soil”.
- Line 114-129: This information should be included in Table-1. In Table-1 Location of each plots and crown position (West x East..) are not important information for enzyme activities and also repeated list of understory species should be summarized.
- Are the forest sites plantation forests? Then Please mention plantation history.
- Please consider to include a site sketch map.
- Line 117: The authors selected herb, shrub, mono and mixed-cultured forest plantations. It is not appropriate to designate a plot on the basis of dominant plant species, Because for example if in a plot four types plant species (A=4 plants, B=3, C=2, D=3), so A is dominant but other species are together eight! My opinion is vegetation types i.e. herb, shrub plots. etc can be used (Q forest= Quercus plot, Q.P forest= mixed plantation etc.).
- Line 131: Total 225 samples (5 sampling sites x 3 replicated plots x 3 (30 x 30m) x …., please explain sample number. How soil samples were collected (using…)?
- Line 177: “ATAN2” please explain what it for.
Results
- Table-1 Mixed forests (QP 1,2,3) are less biodiverse than single forest (Q123, P123)!
- Line 191-205: Describe just the highest and the lowest values of all soil parameters is not proper way of results, rather overall what was the combined effects of different parameters on a particular vegetation types should be mentioned precisely.
- Line 219-226, Line 260.267: Similarly, results should not mean the “language version of the graph/figure” . Which can be seen in graph not necessary to describe in language.
- Same explanation of C, P, Q, QP etc was repeated in the captions of all figures, this should be removed instead of “C.: herb dominated by Cymbopogon goeringii” simply “Herb” can be used in graphs and captions. Captions should not be the explanation of graph title. Line 209: “NH4 + -N stand for soil ammonia nitrogen” this is useless, because it is clear from graph.
- What is the “violin” in graphs -please explain in captions.
Discussions:
- Line 289: grass (?)-shrub-tree
- Line 296: I am not sure moderately dry period affect plant photosynthesis or not.
- Line 300: I didn’t find any Table-2 in the manuscript.
- Line 399: “Hu’s [34] research showed that the soil pH of different forest types in natural secondary forests and primary forests was 5.08-5.54, and the SOC was 47.29-124.38 mg·g-1” Is this exactly same location of current study? If not please don’t compare soil moisture without comparing soil texture.
- Line 302-308: Comparisn soil properties from literature needs careful checking of the both locations for similar background data.
- Line 315-320: Literature review should be removed to introduction section.
- Line 320-326 : This is just repetition of the results, please delete this.
- Line 372-375: Repetition of Materials and Methods.
- Line 415: “ecological enzymes by affecting the activity of extracellular enzymes” what is the difference between “ecological enzymes” and “extracellular enzymes”
Author Response
Responses to Reviewers 4 Comments
MS Type: Article
Title: Extracellular enzyme stoichiometry reveals soil microbial carbon and phosphorus limitations in the Yimeng Mountain area, China
Dear Revision Quality Check:
Thank you very much for your attention and the referees’ evaluation and comments on our paper. These comments were very valuable and helpful for revising and improving our article, as well as guiding the significance of our research. We have studied the comments carefully and have made corrections in the manuscript, and the responses to the reviewers’ comments are as follows.
- Line 41-43: “The level…. but also characterize …..but its role is also…” Please split the sentence.
Responses: We have revised the sentence, “The level of extracellular enzymes activity can reflect the functional traits of soil microorganisms, and it characterize the potential of soil for specific biochemical reactions, which role are also limited by various environmental factors”.
- Line 45: “…organic carbon also acts on a variety of microorganisms to promote enzyme activity” the sentence doesn’t mean any strong sense.
Responses: We have deleted it.
- In entire manuscript the word “ecological” “eco- enzymatic” excessively and sometime unnecessarily, for example: “soil ecological enzyme activity (Line 48)”, Line 50-52: “Ecoenzymatic stoichiometry (is it differ if we say Soil enzymatic stoichiometry?) theory combines ecological metabolism theory and ecological stoichiometry theory to estimate the limitations of ecosystem microbial metabolism from the perspective of enzyme activity”. I think the whole manuscript needs to revise for such excess words.
Responses: We have uniformly revised “ecological enzymes”, “ecological enzymes activity” and “soil enzymes” as “soil extracellular enzyme” and deleted excess words.
- Line-66: “Therefore……. different environmental conditions is very …. Ecosystems”. Irrelevant because environmental condition was not considered here.
Responses: We have revised.
- Line 72-73: Need reference.
Responses: We have added the reference.
- Line 96: “In this study, the soil of five communities was studied” What does mean by “soil of five communities”? Sentence structure was not correct, plural noun and verb and repeated use of study.
Responses: We have rewrite the sentence. “In this study, we investigated the nutrient limitation of microbial metabolisms of the different plots in the state-owned Dawa forest farm in the Yimeng Mountain area, China.”
- Line 96-101: should be transfer to Materials & methods section.
Responses: We have deleted this part and stated in materials and methods section.
- Line 110: “The mean annual temperature of this region is 12.8 °C, the mean annual precipitation is 600 mm, and the area has a warm temperate continental monsoon climate and a frost-free period of 191 d.” Please make it short avoid repeated words!
Responses: We have revised to “The mean annual temperature and precipitation of this region is 12.8 °C and 600 mm, the area has a warm temperate continental monsoon climate and a frost-free period of 191 d, mean annual temperature and precipitation were 12.8 °C and 600 mm.”
- Line 114 &118: “arbor soil” is not generally used, better is “forest soil”.
Responses: We have revised the “arbor soil”.
- Line 114-129: This information should be included in Table-1. In Table-1 Location of each plots and crown position (West x East..) are not important information for enzyme activities and also repeated list of understory species should be summarized.
Responses: Location of each plots and crown position (West x East.) have deleted, and the understory species have been summarized in study site.
- Are the forest sites plantation forests? Then Please mention plantation history.
Responses: The forest sites are plantation forests, and we have supplemented the plantation history.
- Please consider to include a site sketch map.
Responses: We have added a map of the study site to the supplementary material.
- Line 117: The authors selected herb, shrub, mono and mixed-cultured forest plantations. It is not appropriate to designate a plot on the basis of dominant plant species, Because for example if in a plot four types plant species (A=4 plants, B=3, C=2, D=3), so A is dominant but other species are together eight! My opinion is vegetation types i.e. herb, shrub plots. etc can be used (Q forest= Quercus plot, Q.P forest= mixed plantation etc.).
Responses: The plots's name were revised the herb plot, shrub plot, Quercus acutissima plot, Pinus thunbergii plot, and mixed plantation plot.
- Line 131: Total 225 samples (5 sampling sites x 3 replicated plots x 3 (30 x 30m) x …., please explain sample number. How soil samples were collected (using…)?
Responses: We have revised to “There were 225 random points from different sites were collected, using three plots as three independent replicates. In total, 15 sample number were established.”
- Line 177: “ATAN2” please explain what it for.
Responses: We have explained each variable in the formula. “ATAN2” is arctangent.
- Table-1 Mixed forests (QP 1,2,3) are less biodiverse than single forest (Q123, P123)!
Responses: We have deleted this part.
- Line 191-205: Describe just the highest and the lowest values of all soil parameters is not proper way of results, rather overall what was the combined effects of different parameters on a particular vegetation types should be mentioned precisely.
Responses: We have re-analyzed and summarized in-depth of results.
- Line 219-226, Line 260.267: Similarly, results should not mean the “language version of the graph/figure” . Which can be seen in graph not necessary to describe in language.
Responses: We have re-analyzed and summarized.
- Same explanation of C, P, Q, QP etc was repeated in the captions of all figures, this should be removed instead of “C.: herb dominated by Cymbopogon goeringii” simply “Herb” can be used in graphs and captions. Captions should not be the explanation of graph title. Line 209: “NH4+-N stand for soil ammonia nitrogen” this is useless, because it is clear from graph.
Responses: We have revised the explanation. “Figure 1 Descriptive statistics of soil and litter properties, TClitter stand for litter total carbon, TNlitter stand for litter total nitrogen, TPlitter stand for litter total phosphorus. Values are the means ± standard error. The different letters indicate significant difference between treatments according to the Tukey test, lowercase letters (such as a, b, c, d) indicate significant difference (P < 0.05) among the five sampling plots.”
- What is the “violin” in graphs -please explain in captions.
Responses: Figure has been redrawn and the shape of the figure has explained below the figures.
- Line 289: grass (?)-shrub-tree.
Responses: We have revised to “herb to shrub to forests stage.”
- Line 296: I am not sure moderately dry period affect plant photosynthesis or not.
Responses: We have revised and re-analyzed in discussion.
- Line 300: I didn’t find any Table-2 in the manuscript.
Responses: We have revised.
- Line 399: “Hu’s [34] research showed that the soil pH of different forest types in natural secondary forests and primary forests was 5.08-5.54, and the SOC was 47.29-124.38 mg·g-1” Is this exactly same location of current study? If not please don’t compare soil moisture without comparing soil texture.
Responses: We have revised and re-analyzed.
- Line 302-308: Comparisn soil properties from literature needs careful checking of the both locations for similar background data.
Responses: All discussion sections have been re-analyzed in-depth.
- Line 315-320: Literature review should be removed to introduction section.
Responses: We have deleted this part and removed to introduction section.
- Line 320-326 : This is just repetition of the results, please delete this.
Responses: We have deleted this part and re-analyzed in discussion.
- Line 372-375: Repetition of Materials and Methods.
Responses: We have deleted this part and re-analyzed in discussion.
- Line 415: “ecological enzymes by affecting the activity of extracellular enzymes” what is the difference between “ecological enzymes” and “extracellular enzymes”
Responses: The“ecological enzymes” and “extracellular enzymes” are the same. We have uniformly revised “ecological enzymes”, “ecological enzymes activity” and “soil enzymes” as “soil extracellular enzyme”, and have modified this sentence to " Therefore, soil moisture, soil organic C and available P, litter total C and N can indirectly affect the extracellular enzymes stoichiometric and vector characteristics by affecting the extracellular enzymes activities, and thus affecting the nutritional limitation of microbial metabolism."
We tried our best to improve the manuscript and made revised in the manuscript. These changes did not influence the content or framework of the paper. Here, we did not list the changes but marked them in red in our revised paper.
We appreciate the hard work of the editor and reviewers and hope that the corrections will meet with your approval.
Once again, thank you very much for your comments and suggestions.

Round 2
Reviewer 4 Report
The manuscript is technically sound and interesting. But there are many english language errors. So I would recommand for an extensive editing of english language.
For example poor English style: Line 26-28 ( In this study........)
Grammatical mistake: Line 38: (There were significant......)
Author Response
We asked professionals to correct the English mistakes in the manuscript. The editing certificate is attached in the cover letter. But we can't list all the changes here. So we marked the changes in the manuscript in blue.
